# Mediating the Effects of Climate on the Temperature and Thermal Structure of a Monomictic Reservoir through Use of Hydraulic Facilities

Maurice Alfonso Duka [1,2], Tetsuya Shintani [1] and Katsuhide Yokoyama [1,*]

[1] Department of Civil and Environmental Engineering, Tokyo Metropolitan University, 1-1, Minami-Osawa, Hachioji, Tokyo 192-0397, Japan; mauriceduka@gmail.com (M.A.D.); shintani@tmu.ac.jp (T.S.)

[2] Land and Water Resources Engineering Division, IABE, CEAT, University of the Philippines Los Baños, Laguna 4031, Philippines

* Correspondence: k-yoko@tmu.ac.jp; Tel./Fax: +81-42-677-2786

**Abstract:** Climate warming can alter the thermal conditions of reservoirs. However, some hydraulic interventions can be explored to mitigate this impact. This study investigates the long-term effects of climate on the temperature and thermal structure of a monomictic reservoir that has had varying operations from 1959 to 2016. Reservoir progressively operated through three distinct periods, namely, (A) deep penstock withdrawal (DPW; 1959–1991), (B) purely selective withdrawal (SW; 1992–2001), and (C) combination of SW and vertical curtain (VC; 2002–2016). Although annual air temperatures are increasing (+0.15 °C decade$^{-1}$) in the long term, the reservoir's surface water temperatures have been found to be decreasing (−0.06 °C decade$^{-1}$). Periods B and C produced colder profiles and exhibited lower heat content and higher potential energy anomaly than Period A. Furthermore, stronger thermoclines, as indicated by Brunt–Vaisala frequency, were observed in the two latter periods. The results of this study show that varying operations bear a stronger influence on the reservoir's temperature and thermal structure than climate change itself. Mitigating the thermal impacts of climate warming in reservoirs appears promising with the use of SW and VC.

**Keywords:** surface water cooling; reservoir cooling; climate warming; selective withdrawal; vertical curtain; deep penstock withdrawal

## 1. Introduction

Climate warming, as typically indicated by a rise in air temperature, poses serious impacts on the thermal condition of inland water bodies such as lakes and reservoirs. In terms of surface water temperatures (SWTs), for example, long-term warming trends in lakes have been found to be directly associated with rising trends in global air temperature [1–4]. Similar findings were detailed in the studies of Ficker et al. [5], Schneider and Hook [6], O'Reilly et al. [7], Woolway et al. [8], and Woolway and Merchant [9] for various lakes worldwide. The rise in SWT can result in (1) increased rates of evaporation [10,11], (2) elevated rates of bacterial and phytoplankton activities [8], (3) enhanced thermal resistance to vertical mixing [12], (4) shorter period of ice cover, specifically for dimictic lakes [1], and (5) proliferation and invasion of warm-water aquatic species [11]. Because of their direct response to climate forcing, lakes are identified as sentinels of global warming [13].

A number of recent studies have investigated the responses of reservoirs to climate change. For example, Lake Dillon in Colorado, USA, exhibited a notable increase in SWT, heat budget, and stability due to climate warming [14]. In Lake Qiandahou in China, the increase in air temperature was associated with stronger dissolved oxygen stratification and a decrease in oxycline depth [15]. Several studies have also used 2D modeling through CE-QUAL-W2 (Portland State University, USA) to evaluate the effect of climate warming on various reservoirs. For instance, the hypereutrophic Hodges Reservoir in the USA is

projected to experience an increase in evaporation rates, stronger stratification, and overall water column warming [16]. The Aidoghmoush Reservoir in Iran could be subject not only to an increase in both surface and bottom water temperatures but also in its total dissolved solids (TDSs) [17]. The water quality of the Hsinshan Reservoir in Taiwan is also projected to deteriorate, with a reduction of dissolved oxygen concentrations at the bottom layer and an increase in phosphorus concentration [18]. Interventions are, therefore, needed to mitigate the effects of climate warming on reservoirs as they are significant sources of fresh water in the world.

Reservoirs behave differently compared to natural lakes. While lakes are typically restricted by surface outflows, reservoirs can deliberately discharge at several locations throughout their depth [19]. Lakes also characteristically have lower flushing rates and longer hydraulic retention periods than reservoirs [20]. The two types of inland water bodies, therefore, generally have considerably different temperature dynamics and respond differently to climate. In fact, one study revealed that the management operation of a reservoir with multiple outlets is considered to be the main driver of the thermal conditions of this water body [21]. Another study also showed that dynamic withdrawals in reservoirs can potentially mitigate the effects of climate change [22]. Furthermore, reservoirs are predicted from a conceptual model to have a robust capacity in mediating the effects of climate, especially when some water management practices are implemented [20]. One type of these management filters involves the use of variable withdrawal techniques and the installation of hydraulic facilities that can modify the thermal structure of the reservoir [23]. Nowadays, selective withdrawal (SW) systems can be retrofitted in reservoirs that allow warm water releases from the epilimnion, while vertical curtains (VCs) can be installed in the upstream reaches to control the direct inflow of river water into the main reservoir body. Determining the combined effects of these two facilities on the reservoir's thermal regime is particularly of high interest, as most available literature deal only with their individual functions.

SW systems are installed primarily to address the problem of cold-water pollution. Conventional hydropower dams usually abstract water from the reservoir's deeper layer (hypolimnion), and this process enables the release of cold water, which is detrimental to the downstream aquatic ecosystem [24]. The installation of SW systems enables the release of water from the warm layers of the reservoir to counter the effects of cold-water pollution and avoid the low dissolved oxygen concentrations downstream. The operation of an SW facility in reservoirs needs to be optimized so that suitable water temperatures are maintained downstream [25,26] not only during summer but also during cold seasons. In terms of the thermal structure of the reservoir, several studies that have employed numerical simulations point out that surface releases through the SW system would increase the thermal stability of the reservoir while bottom releases would induce the warming of the entire water column [27–29]. One example of an SW facility is a retrofit to an existing water supply and hydropower dam in the Ogouchi Reservoir in Japan, allowing epilimnetic releases only [23]. The function of this facility is not limited to regulating outflow temperatures; it also releases highly turbid water in the reservoir during times of flood [30]. Long-term records of water temperature profiles are available for this reservoir; these records can be utilized to evaluate and compare the actual reservoir's thermal responses when hypolimnetic withdrawals are made by penstock or through epilimnetic outflows by SW.

VCs, on the other hand, are structures installed across the river mouths to prevent the direct intrusion of river water into the reservoir [31]. A VC can also be installed in other sections of the reservoirs than the river mouths. It has been called many names, such as vertical weir curtain [32], floating curtain weirs [33], flexible curtain [34], and temperature control device [35], among others. Some studies have discussed the effectiveness of the curtain for regulating outflow temperature for fishery purposes [36], controlling algal blooms [31,33], and mitigating the occurrence of cyanobacteria and metabolites [32]. In the Ogouchi Reservoir, the curtains are installed across river mouths, and it was found

that using them aided in lowering the SWT of the reservoir's upstream section [37] and modifying the temperature and velocity distributions before and after the VC [38]. However, the mechanism of this surface cooling phenomenon due to the VC and the effect of combining the VC's operation with SW to the reservoir's thermal structure have yet to be further elaborated.

The temperature and thermal structure of the reservoir govern turbidity current dynamics and phytoplankton movement, which are greatly related to the sedimentation and eutrophication of the water body [39] and its overall water quality status [40]. Therefore, understanding the factors affecting the thermal conditions of the reservoir, such as climate and reservoir operation, is key to water quality management. In this study, the influence of climate on the temperature and thermal structure of a reservoir that has undergone varying operation schemes is investigated. The Ogouchi Reservoir in Japan was chosen as the study site as it had apparently experienced climate warming between 1959 and 2016, and its operation had transitioned through three distinct periods within this duration with the use of deep penstock withdrawal (DPW), purely SW, and a combination of SW and VC [23]. This paper attempts to answer how these different facilities affect the thermal condition of the reservoir and what roles they play as climate warming happens in the reservoir. A wealth of well-documented and comprehensive data is available for the study area, which can robustly support the goals of this study by focusing on the analysis of actual long-term data rather than relying solely on climate projections and numerical simulations.

## 2. Materials and Methods

### 2.1. Site Background

The Ogouchi catchment (Figure 1a), with an area of 263 km$^2$, has six weather observation stations with more than 50 years of meteorological records. Three tributaries drain towards the reservoir, and each has its own gauging station with almost the same length of records as the weather stations. Figure 1b provides the reservoir bathymetry, where the observation point for temperature and other water quality parameters is indicated. Additionally, Figure 1c shows the location of the facilities, such as dam penstock, SW facility, and VC, along the longitudinal section. This study points out three distinct periods of operation, namely, (1) Period A for DPW operation for 1959–1991, (2) Period B for a purely SW operation for 1992–2001, and, lastly, (3) Period C for the combined operation of SW and VC for 2002–2016. More comprehensive details of the study site and the reservoir's operation history can be referred to in the paper of Duka et al. [23].

### 2.2. Climate Analysis

Daily values of air temperature and wind speed were obtained from the dam station, while the rainfall values came from six observation stations (Table 1). While the dam station contains all three meteorological parameters, the other five stations only have precipitation records. These data are from the Automated Meteorological Data Acquisition System (AMeDAS), which is managed by the Japan Meteorological Agency. The recent values of these climatic drivers, from 2012 to 2016 (Figure 2) from the dam station, were examined to highlight their typical daily variations, which were later on used as a basis to identify the two seasons in a year, namely, summer half-year and winter half-year. Then, air temperatures and wind speeds were analyzed based on their annual averages and the averages for the two seasons using the dam station data; for rainfall, only the totals of the annual event and extreme events (>50 mm d$^{-1}$) were analyzed for their basin averages using all six stations.

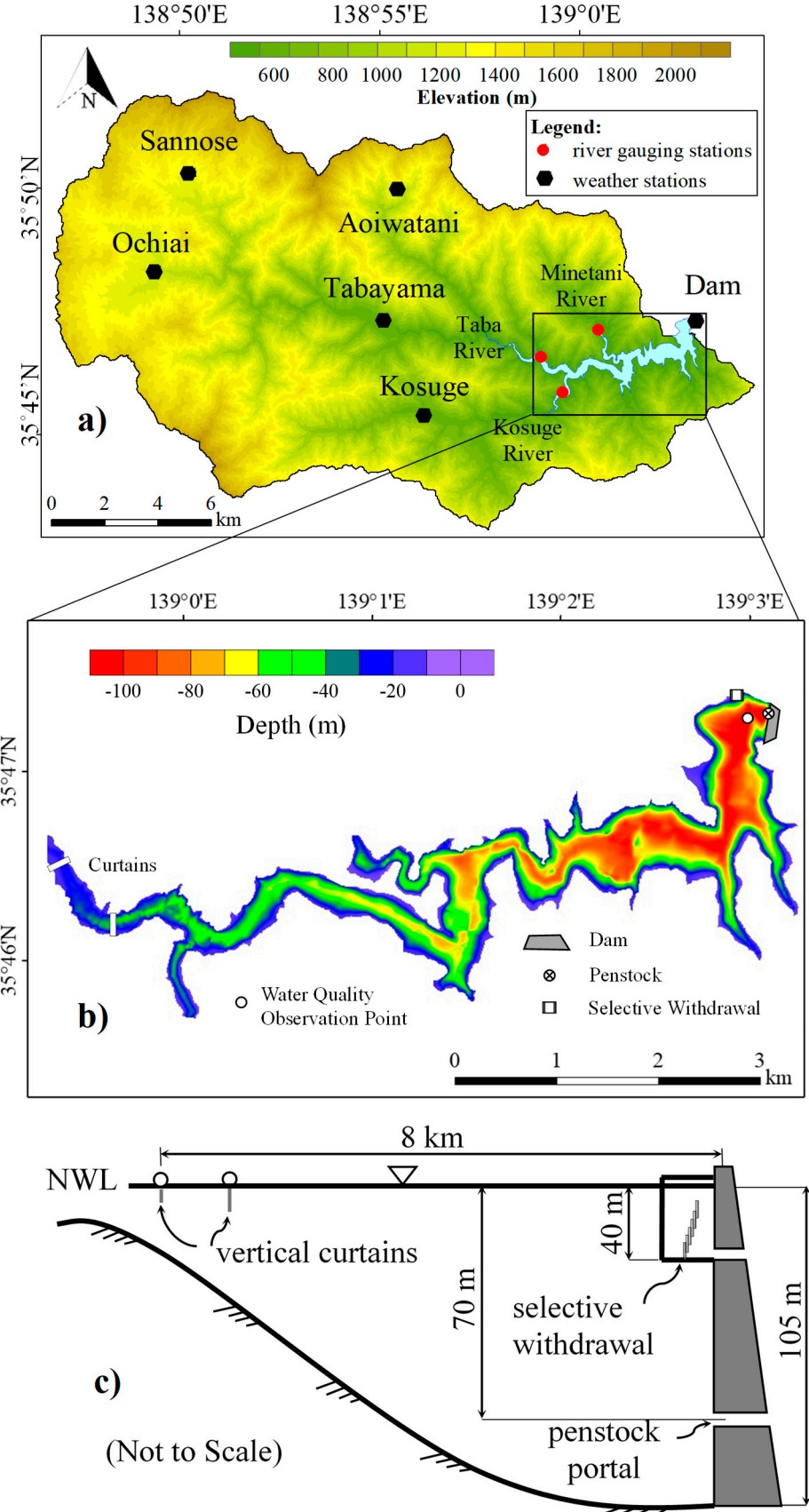

**Figure 1.** Maps of (**a**) the Ogouchi catchment, (**b**) reservoir bathymetry, and (**c**) dam and reservoir profile.

**Table 1.** Long-term observational data of climate and reservoir temperatures.

| Parameters | Station | Elevation (above msl) | Period Covered | Frequency |
|---|---|---|---|---|
| Air temperature (°C) | Dam | 519 m | 1959–2016 | Daily |
| Precipitation (mm) | (1) Dam | 519 m | 1959–2016 | Daily |
| | (2) Ochiai | 1113 m | 1959–2016 | Daily |
| | (3) Tabayama | 611 m | 1959–2016 | Daily |
| | (4) Kosuge | 656 m | 1959–2016 | Daily |
| | (5) Aoiwatani | 1217 m | 1965–2016 | Daily |
| | (6) Sannose | 1268 m | 1965–2016 | Daily |
| Wind Speed (m s$^{-1}$) | Dam | 519 m | 1977–2016 | Daily |
| Water temperature profile (°C) | Upstream of Dam Wall | | 1959–2001 | Weekly |
| | | | 2003–2016 | Daily |

The annual average air temperature in the Ogouchi Reservoir is 15.4 °C, with a maximum of 36.3 °C and a minimum of −12.8 °C between 1959 and 2016. The daily variation in air temperature (Figure 2a) shows that peaks are prominent in August and the lows in January. A typical year can be initially subdivided into a summer half-year (April to September) and a winter half-year (October to March). The summer half-year is classified when the air temperature is equal to or greater than the annual average air temperature, while the winter half-year is when temperatures are below it. Figure 2b shows the typical daily rainfall pattern, wherein the rainy season onsets in the late spring while the typhoon season, bringing large amounts of rainfall, occurs in summer and autumn. Figure 2c provides the recent typical daily wind speed, where cold winds are particularly strong from November to March due to the prevalence of northwesterly winds [41]. Interestingly, strong rain events are mostly concentrated during the summer half-year while stronger winds occur during the winter half-year. This further justifies why this paper adopts the two specific seasons in analyzing the interaction of climate with the thermal conditions of the reservoir.

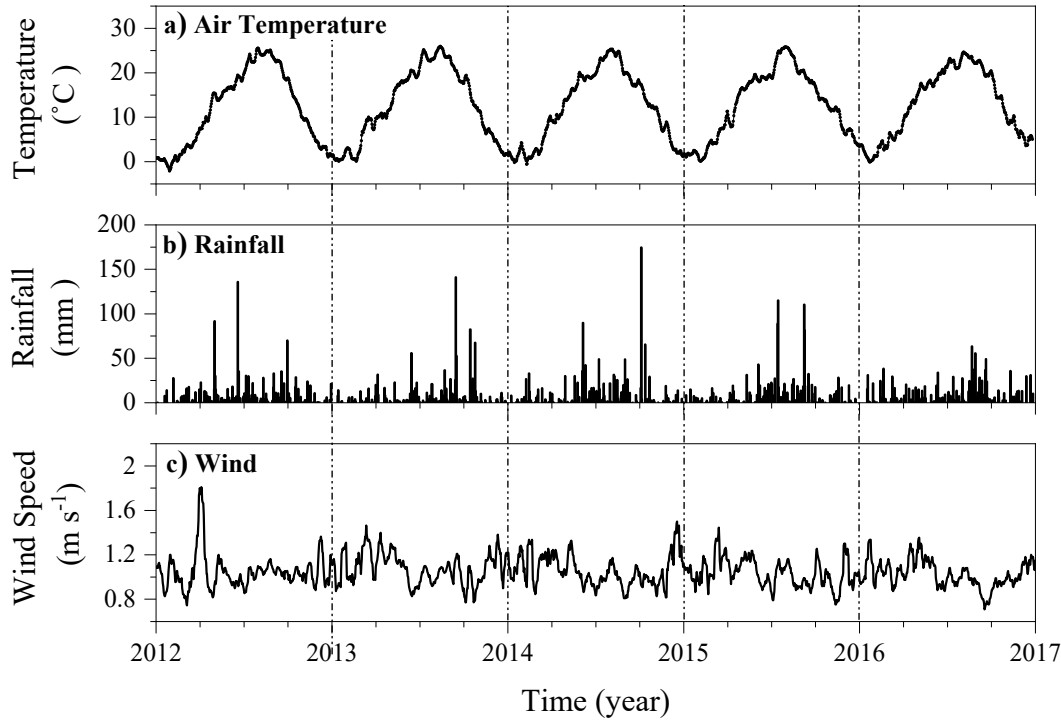

**Figure 2.** Typical variation of (**a**) air temperature (11-day moving average), (**b**) rainfall, and (**c**) wind speed from 2012 to 2016 from the dam station.

Long-term trends were evaluated using the Mann–Kendall (M–K) test [42,43] at a significance level α of 0.05. A nonparametric estimate of the slope of the trend, called Sen's slope [44], is used in this study rather than that from regression based on the least-squares method. The M–K test was performed for the long-term data; henceforth, the magnitude of the slope is expressed in terms of Sen's slope. For short-term trends, i.e., for every period (A, B, and C), simple linear regression was carried out to determine the slopes of meteorological parameters. Since the M–K test was not carried out for short-term trends, the slope was only expressed using linear regression for simpler analysis.

*2.3. Reservoir Temperatures*

Weekly monitoring records of reservoir temperatures are available from 1959 until 2001, while daily records can be accessed from 2003 until 2016 (Table 1) from the Bureau of Waterworks of the Tokyo Metropolitan Government [23]. Raw temperature values from depths of 0, 10, 20, 30, and 70 m were examined. The temperatures at these depths were averaged for the year and the two seasons. The long-term trends of SWT were subjected to the M–K test and the subsequent results were compared with the trends in air temperature. Differences among the three periods of operation were evaluated using Kruskal–Wallis and rank-sum tests. It has to be clarified that the analysis in this paper consists only of a single measuring station for water temperature. The availability of remote-sensed temperature data for the free surface [45] could be useful to shed light on the temperature distribution of the upper layer of the reservoir and establish the roles played by the tributaries.

*2.4. Thermal Structure Indicators*

To quantitatively define the thermal structure of the reservoir, this paper uses three parameters, namely, heat content (Q), potential energy anomaly (PEA), and Brunt–Vaisala Frequency ($N^2$).

The formula for heat content with units of J is

$$Q = \sum_{z_0}^{z_{max}} m c_v T \tag{1}$$

where $m$ is the mass (kg) of water at each defined layer z, $c_v$ is the specific heat of water (4200 J/kg−K), and $T$ (°C) is the water temperature [21].

PEA [46] with units of J m$^{-3}$, on the other hand, is calculated as

$$PEA = \frac{1}{H} \int_{-H}^{0} (\bar{\rho} - \rho) g z \, dz \tag{2}$$

where $H$ (m) is the total depth of the reservoir, $z$ (m) is the depth of water from the surface, $\rho$ (kg m$^{-3}$) is water density at a certain layer and temperature, $\bar{\rho}$ (kg m$^{-3}$) is the vertically averaged potential density, and g (9.80 m s$^{-2}$) is the acceleration due to gravity. PEA as a measure of "difficulty in mixing" indicates the amount of energy needed to vertically homogenize the water column [47].

Lastly, $N^2$ (s$^{-2}$) is given by

$$N^2 = -\frac{g}{\rho} \frac{\partial \rho}{\partial z} \tag{3}$$

as a measure of the strength of the buoyancy [48]. The weekly values of $N^2$ were computed for each layer from the surface to the bottom of the reservoir for all available data from 1959 to 2016. Contour plots were made to represent average stability in a year for each of the three periods.

## 3. Results

*3.1. Climate Trends*

Results of the M–K test are provided in Table 2, where significant upward trends in air temperature are observed for the year and winter half-year at +0.15 and +0.30 °C decade$^{-1}$,

respectively. The annual air temperature rise is consistent with the +0.12 °C decade$^{-1}$ officially recorded increase in Japan between 1898 and 2016 [49]. The increase in air temperature during the winter half-year corresponds with the warmer winter and autumn, as evidenced by the documented rise of +0.11 and +0.13 °C decade$^{-1}$, respectively. For the Tokyo metropolitan area, where the catchment belongs, a study [50] revealed that the annual mean temperature increased by about +0.30 °C decade$^{-1}$ between 1901 and 2015, a value way larger than the national average of +0.12 °C decade$^{-1}$ [49]. This could explain the higher rate of increase during the winter half-year for the Ogouchi catchment. However, no significant trend in air temperature could be detected in the summer half-year, which means that the annual trend is largely affected by the rising local temperatures during the winter half-year. On the other hand, wind speeds showed a nonsignificant upward trend for the annual and the two seasons. Furthermore, the long-term trends of annual and heavy rainfall (>50 mm d$^{-1}$) are not statistically significant. While the country's air temperature displayed upward trends, which are likely attributed to climate change and urbanization, precipitation may be considered to be within the normal range of fluctuations [51].

**Table 2.** Results of the Mann–Kendall (M–K) test at $\alpha = 0.05$ for long-term values of air temperature, wind speed, rainfall (basin average), and SWT. Significantly different *p*-values are in bold characters.

| Parameter | Period | M–K z-Stat | M–K *p*-Value | Sen's Slope |
|---|---|---|---|---|
| Air Temperature | Annual | +2.9783 | **0.0029** | +0.15 °C decade$^{-1}$ |
| | Apr–Sept | −0.1610 | 0.8271 | −0.01 °C decade$^{-1}$ |
| | Oct–Mar | +4.4809 | **<0.0001** | +0.30 °C decade$^{-1}$ |
| Wind Speed | Annual | +1.8758 | 0.0607 | +0.03 m s$^{-1}$ decade$^{-1}$ |
| | Apr–Sept | +1.4565 | 0.1453 | +0.02 m s$^{-1}$ decade$^{-1}$ |
| | Oct–Mar | +1.8293 | 0.0673 | +0.03 m s$^{-1}$ decade$^{-1}$ |
| Rainfall | Annual | +0.2147 | 0.8300 | +5.02 mm decade$^{-1}$ |
| | >50 mm d$^{-1}$ | +0.2415 | 0.8092 | +3.35 mm decade$^{-1}$ |
| SWT | Annual | −1.2723 | 0.2033 | −0.06 °C decade$^{-1}$ |
| | Apr–Sept | +0.0566 | 0.9549 | <0.01 °C decade$^{-1}$ |
| | Oct–Mar | −2.3253 | **0.0201** | −0.15 °C decade$^{-1}$ |

The long-term trends in air temperature, wind speed, and basin-averaged rainfall for the year and the two seasons can be referred to in Figure 3. Long-term atmospheric warming during the year and the winter half-year can be confirmed, with overall rising trends in air temperature in Figure 3a,c. Besides these two, no other significant long-term trends were identified for the rest of the parameters (Table 2); however, several remarkable observations can be recognized at certain years. Figure 3b shows that the period from 1976 to 1993 is characterized by cold spells during the summer half-year. In terms of wind speed (Figure 3d–f), recent years from 2007 onwards are dominated by large values. In terms of annual rainfall (Figure 3g), the wettest years (magnitudes greater than 2000 mm) were 1959, 1991, and 1998, while the driest years (below the average of 1480 mm) were 1973, 1980, 1984, and 2009. In Figure 3h, strong rainfall events were observed in 1974, 1983, and 2011, due primarily to typhoons.

The meteorological trends in each period were defined by the slope of the parameters using simple linear regression. For Period A, air temperature, wind speed, and basin-averaged rainfall are characterized by weak slopes. On the other hand, Period B noticeably has rising trends in air temperature and basin-averaged rainfall but a decreasing one in wind speed. A limitation to estimating the trends for Period B is recognized as it covers only a relatively short duration of ten years. Furthermore, Period C manifests a

slight increase in air temperature and a strong increase in wind speed but the reverse for basin-averaged rainfall.

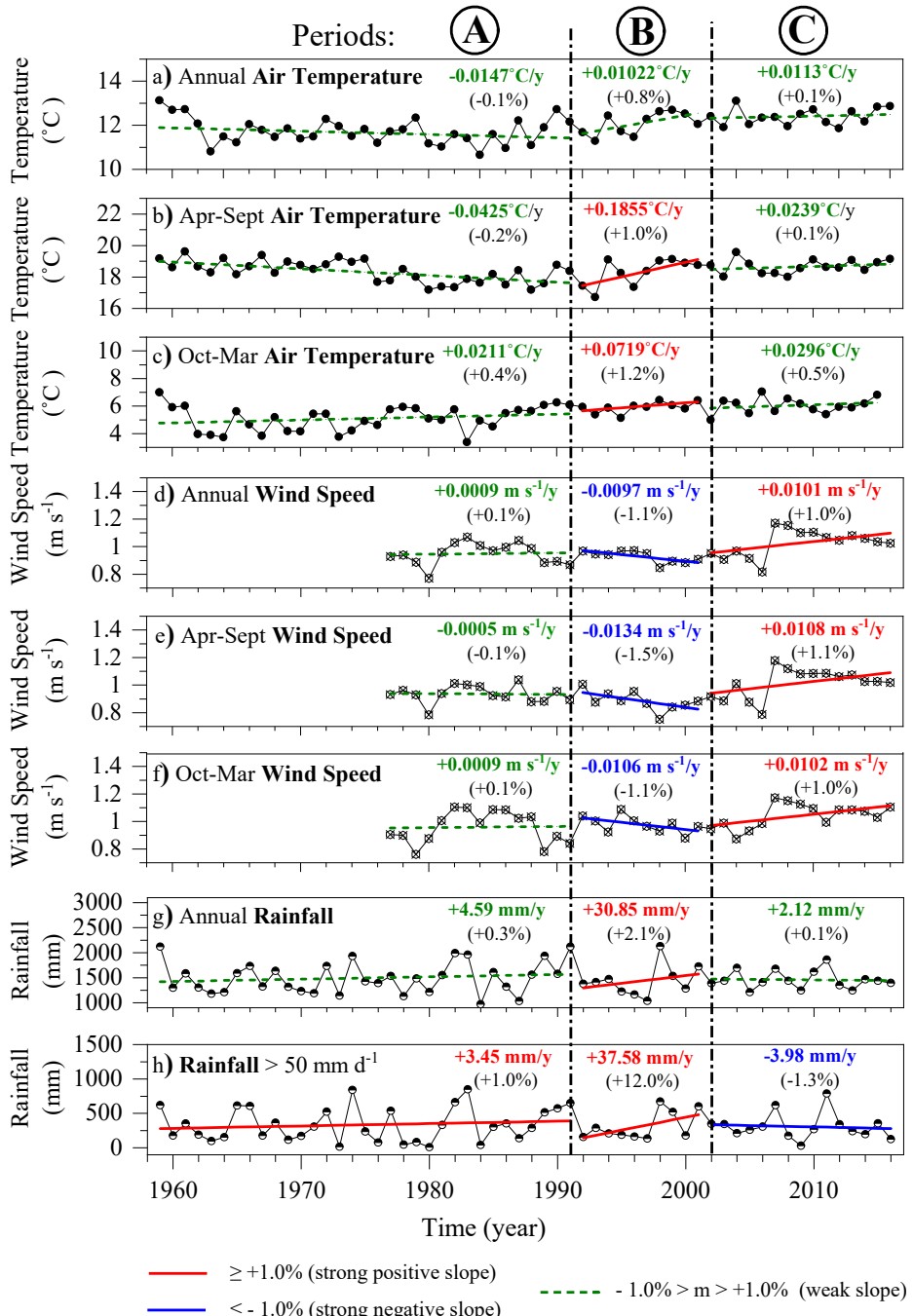

**Figure 3.** Average air temperature (**a–c**), wind speed (**d–f**) and basin-averaged rainfall (**g,h**). Colored straight lines are linear fit for each period, with colored texts as slopes, while values in parentheses are slope percentages (slope value/average value for that period × 100).

## 3.2. Water Temperature Distribution and Trends

Figure 4 presents the raw reservoir temperatures recorded at different depths from 1959 to 2016, while Figure 5 provides the average weekly temperature for the year for each period. The general pattern follows a trend where low values can be observed during winter (December to February), peaking during summer (August) and decreasing with the onset of autumn. Large fluctuations are observed for SWT (0 m), with a high of 25 °C and a low of 5 °C for all three periods. On the other hand, Period A (Figures 4a and 5a) shows warmer

conditions for the 10-, 20-, 30-, and 70-m depths, while Periods B (Figures 4b and 5b) and C (Figures 4c and 5c) are mostly colder at these depths. Temperatures at the 70-m depth for the two latter periods are relatively flat at the 5 °C level. Closer lines between depths mean a relatively mixed condition at these layers, while gaps between the layers indicate an apparent thermocline. For Period A, small temperature gaps are evident between the 30- and 70-m depths from late summer until autumn. Meanwhile, large gaps are manifested for Periods B and C between the 10-m and 30-m depths over the entire year, except during winter, where mixing occurs. Between the 30- and 70-m depths for Periods B and C, temperature gaps are larger for the latter, meaning that Period C exhibits deeper thermocline and thicker epilimnion than Period B.

In terms of heat content (Figure 5d), the Q peaks around Day 270 (end of September), while it is the lowest around Day 75 (mid-March). Period A consistently attained the highest amount of heat stored in the reservoir. Periods B and C have similar trends, although Period C has relatively higher heat content than Period B. Although the SW facility was in operation in both these two latter periods, the effect of VCs may have played a role in the higher heat content of Period C. The observed differences in the plots of temperature profiles and heat content suggest that the operation of the facilities significantly influenced the thermal structure of the reservoir. This is further explained in the succeeding sections.

For SWT (0 m), no observable differences can be seen between the three periods during the summer half-year; however, the two latter periods appear to be generally colder than Period A during the winter half-year (Table 3). On average, the SWT for Periods B and C decreased relative to Period A. Kruskal–Wallis tests showed significant differences among the periods ($p = 0.0019$) for the winter half-year (October to March), while rank-sum tests further confirmed the significant differences of Periods B and C with Period A. Furthermore, based on M–K analysis, the SWT decreased for the year at $-0.06$ °C decade$^{-1}$ (nonsignificant, $p = 0.2033$) and for the winter half-year at $-0.15$ °C decade$^{-1}$ (significant, $p = 0.0201$), with no observable trend for the summer half-year (April to September) (Table 2). During the year and the winter half-year, the SWTs were decreasing, although air temperatures were significantly rising. This surface cooling phenomenon can be strongly attributed to the change in management procedures.

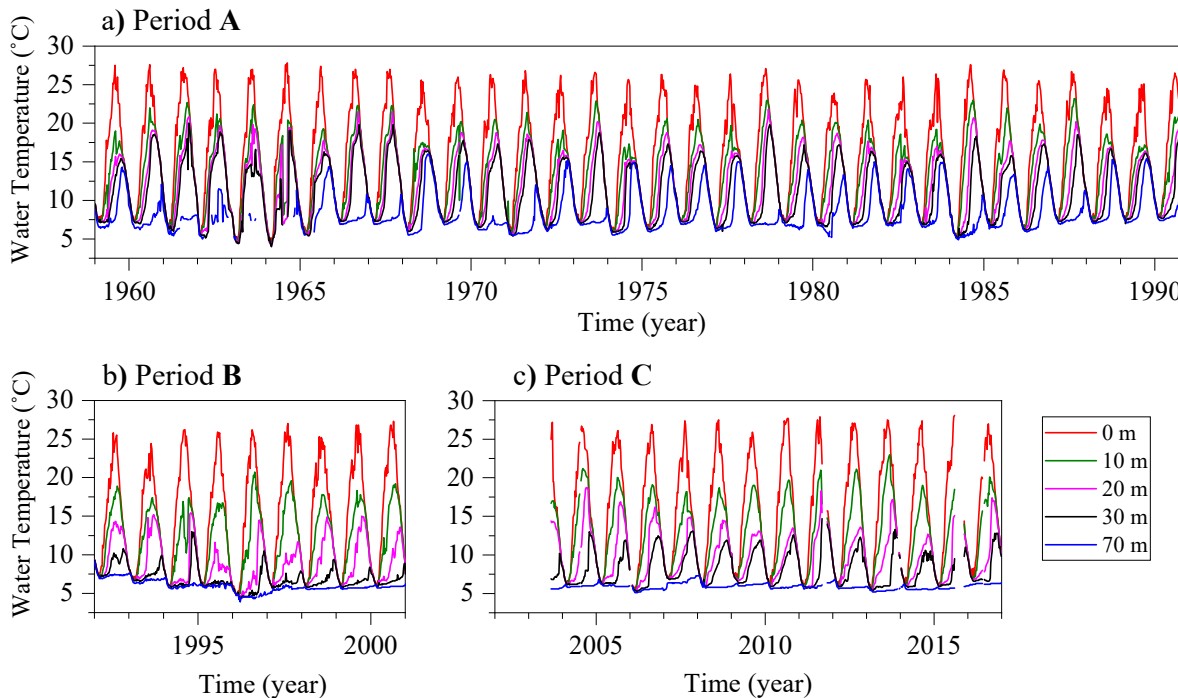

**Figure 4.** Long-term fluctuations in water temperature at different depths for the three periods (**a–c**). Data are unavailable for 2002 and largely incomplete for 2003.

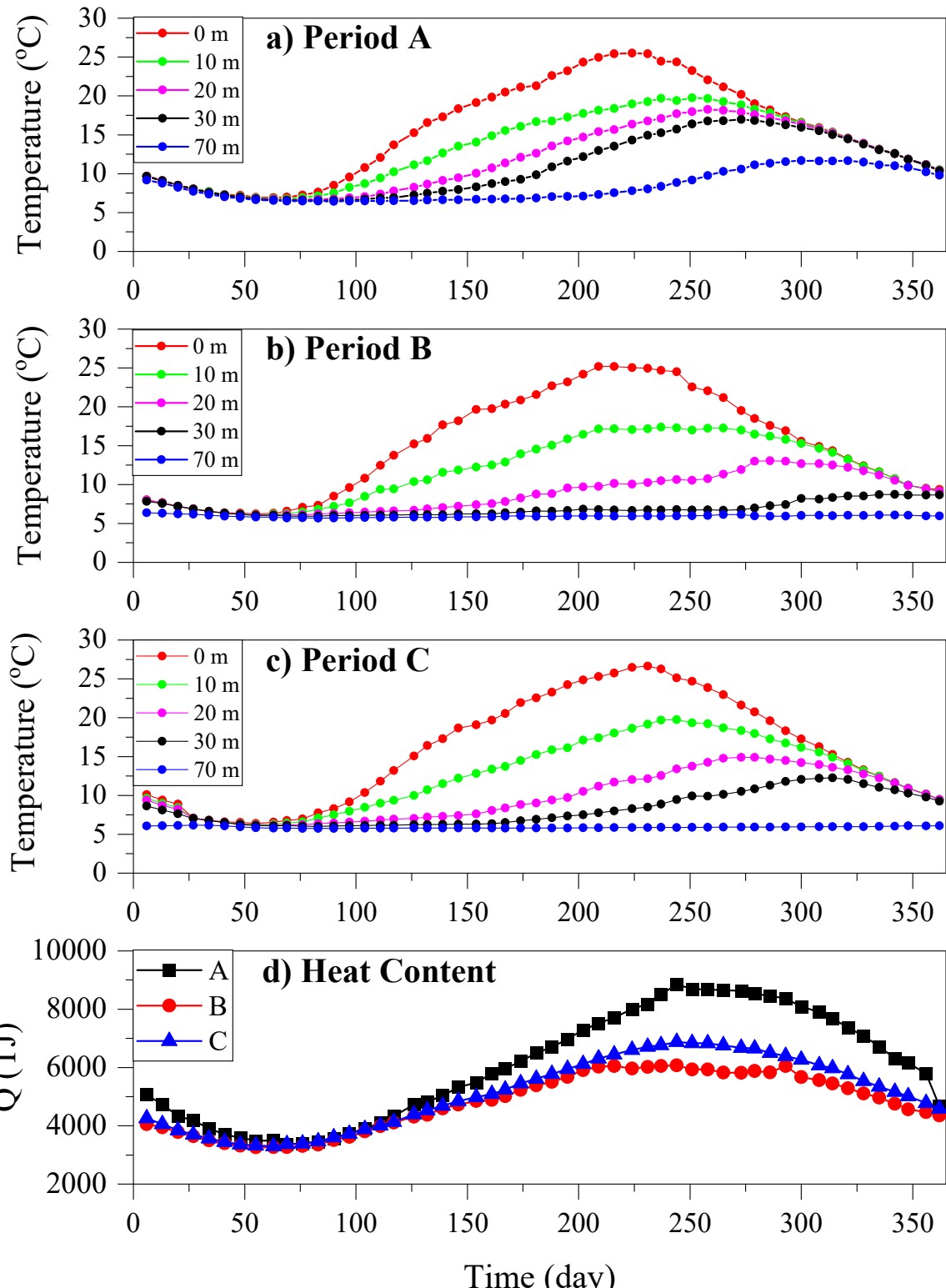

**Figure 5.** Average weekly water temperatures at different depths for different periods (**a**–**c**) with their corresponding heat content (**d**). To get the average per period, the transitional years between two periods are eliminated.

For the temperatures below the water surface (10 to 70 m), the Kruskal-Wallis test affirmed the significant differences among periods. Period A is generally warmer than B and C, on average (Table 3). During the summer half-year, large temperature gaps in the upper layer are observed in Periods B and C, indicating the presence of strong thermoclines

(Figure 6a). During the winter half-year (Figure 6b), while Period A is mostly isothermal, Periods B and C still have inherent stratification, as observed from the relatively large temperature gaps between the 20- and 70-m depths.

**Table 3.** Results of Kruskal–Wallis (K–W) and rank-sum tests at $\alpha = 0.05$ for water temperatures at various depths. Significantly different *p*-values are in bold characters.

| Depth | Season | Average per Period (°C) | | | K–W *p*-Value |
|---|---|---|---|---|---|
| | | A | B | C | |
| 0 m | Apr–Sept | 20.09 | 20.09 | 20.40 | $3.6 \times 10^{-1}$ |
| | Oct–Mar | 11.23 | 10.44 * | 10.73 * | **$1.9 \times 10^{-3}$** |
| 10 m | Apr–Sept | 15.58 | 13.98 * | 14.61 * | **$8.4 \times 10^{-6}$** |
| | Oct–Mar | 11.06 | 10.08 * | 10.22 * | **$1.6 \times 10^{-3}$** |
| 20 m | Apr–Sept | 12.87 | 8.63 * | 9.75 * | **$1.9 \times 10^{-9}$** |
| | Oct–Mar | 10.91 | 9.31 * | 9.47 * | **$4.7 \times 10^{-6}$** |
| 30 m | Apr–Sept | 11.14 | 6.40 * | 7.41 * | **$1.2 \times 10^{-9}$** |
| | Oct–Mar | 10.76 | 7.46 * | 8.61 * | **$1.0 \times 10^{-8}$** |
| 70 m | Apr–Sept | 7.59 | 5.85 * | 5.79 * | **$1.4 \times 10^{-7}$** |
| | Oct–Mar | 9.20 | 5.91 * | 5.97 * | **$7.0 \times 10^{-9}$** |

* Significantly different with Period A by rank-sum test.

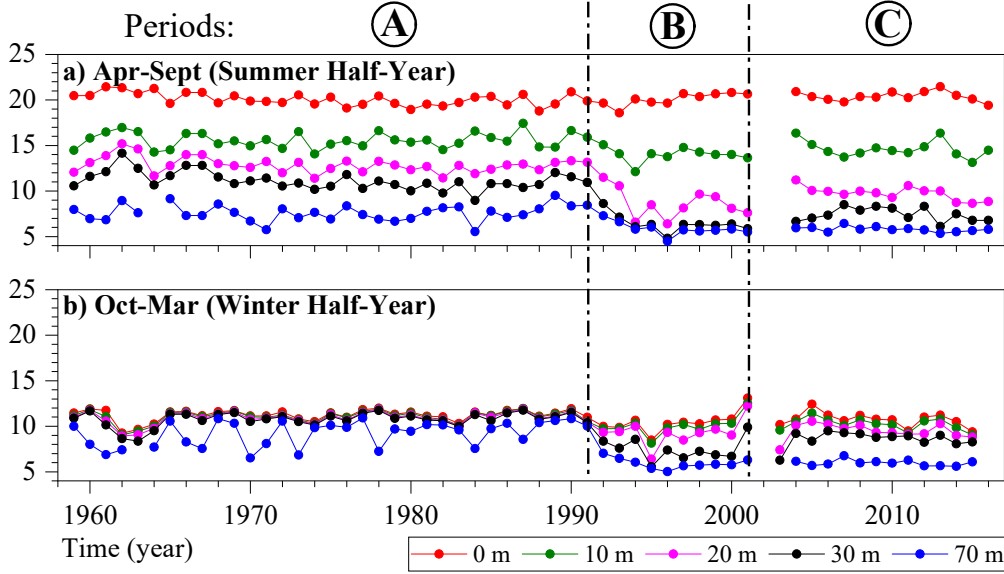

**Figure 6.** Average annual water temperatures at different depths from 1959 to 2016 for (**a**) the summer half-year and (**b**) the winter half-year.

### 3.3. Thermal Structure Parameters

On average, for the summer half-year (Figure 7), Period A shows the highest value of Q, with 6600 TJ, followed by C with 5500 TJ, and, lastly, by B with 5200 TJ. In terms of PEA, Period C obtains the highest average value of 103 J m$^{-3}$, followed by B with 94 J m$^{-3}$ and A with 93 J m$^{-3}$. For the winter half-year, Period A still obtains the highest average Q, followed by C and then B, while stability in terms of PEA is still highest for C, followed by B and then A. In summary, Period A manifests higher Q but lower PEA than Periods B and C. Higher Q is observed for Period A as it has a warmer water column and thicker epilimnion. High stability, as described by PEA, is observed in Periods B and C, as attributed to stronger vertical stratification in these two periods.

The trends of these two thermal structure parameters can also be compared with those of meteorological forcing. Period A, which generally exhibited weak slopes for air temperature and wind speed during both seasons, obtained relatively weak slopes for Q and PEA during the summer half-year but a strong negative slope for PEA during the winter half-year. For Period B during both seasons, the increase in air temperature and rainfall and the decrease in wind speed (Figure 3) are associated with the decrease in Q but the reverse for PEA. On the other hand, for Period C, specifically during the winter half-year, the increase in wind speed is associated with decreasing trends in Q and PEA. To summarize, while the difference in the average values of different parameters of thermal parameters is brought largely by the varying reservoir operations, the trends of these parameters in every period are highly associated with the trends of air temperature and wind speed.

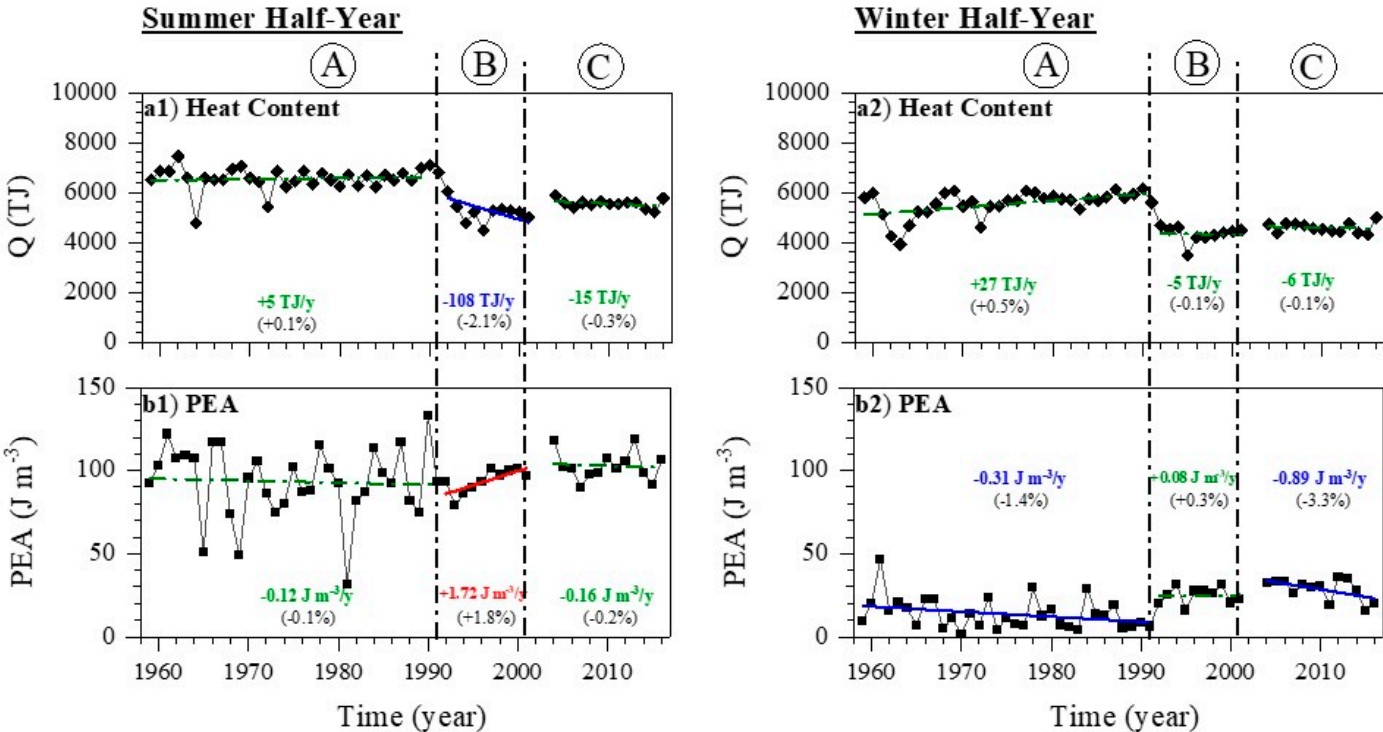

**Figure 7.** Time series plots of different thermal structure parameters such as (**a1**,**a2**) heat content (Q) and (**b1**,**b2**) potential energy anomaly (PEA). Colored straight lines are linear fit for each period, with red representing a strong positive slope (m $\geq$ +1%), blue for a strong negative slope (m $\leq$ −1%), and green for a weak slope (−1% > m > +1%). Colored texts are slopes, while values in parentheses are slope percentages (slope value/average value for that period × 100).

The stability of the reservoir in terms of $N^2$ is shown in Figure 8. For all three periods, the duration between Day 0 (1 January) until around Day 90 (31 March) is characterized by zero values of $N^2$. This indicates that isothermal conditions existed during this specific time range. The $N^2$ values are pronounced from Day 91 (1 April) to Day 273 (30 September), which specifically fall during the summer half-year. From Day 274 (1 October) until the end of the year, the $N^2$ values gradually decrease as the reservoir experiences a weakening of stratification due to overturn during fall and mixing during winter.

Focusing on the summer half-year, stratification is more intense in August for Periods B and C compared to A. Stronger thermoclines were produced during Periods B and C because of the shallow withdrawals through the SW facility. Comparing the two latter periods with the 0.0001 contour line, stratification extends deeper for C due to the effect of the VC.

Period A, on the other hand, has weaker stratification, as attributed to hypolimnetic withdrawals by penstock. Considering the 0.0001 contour line, the stratification extends to

the deepest part of the reservoir, specifically near year-end. This means that thermoclines are developed at the deeper portions of the reservoir during the start of the cooling season.

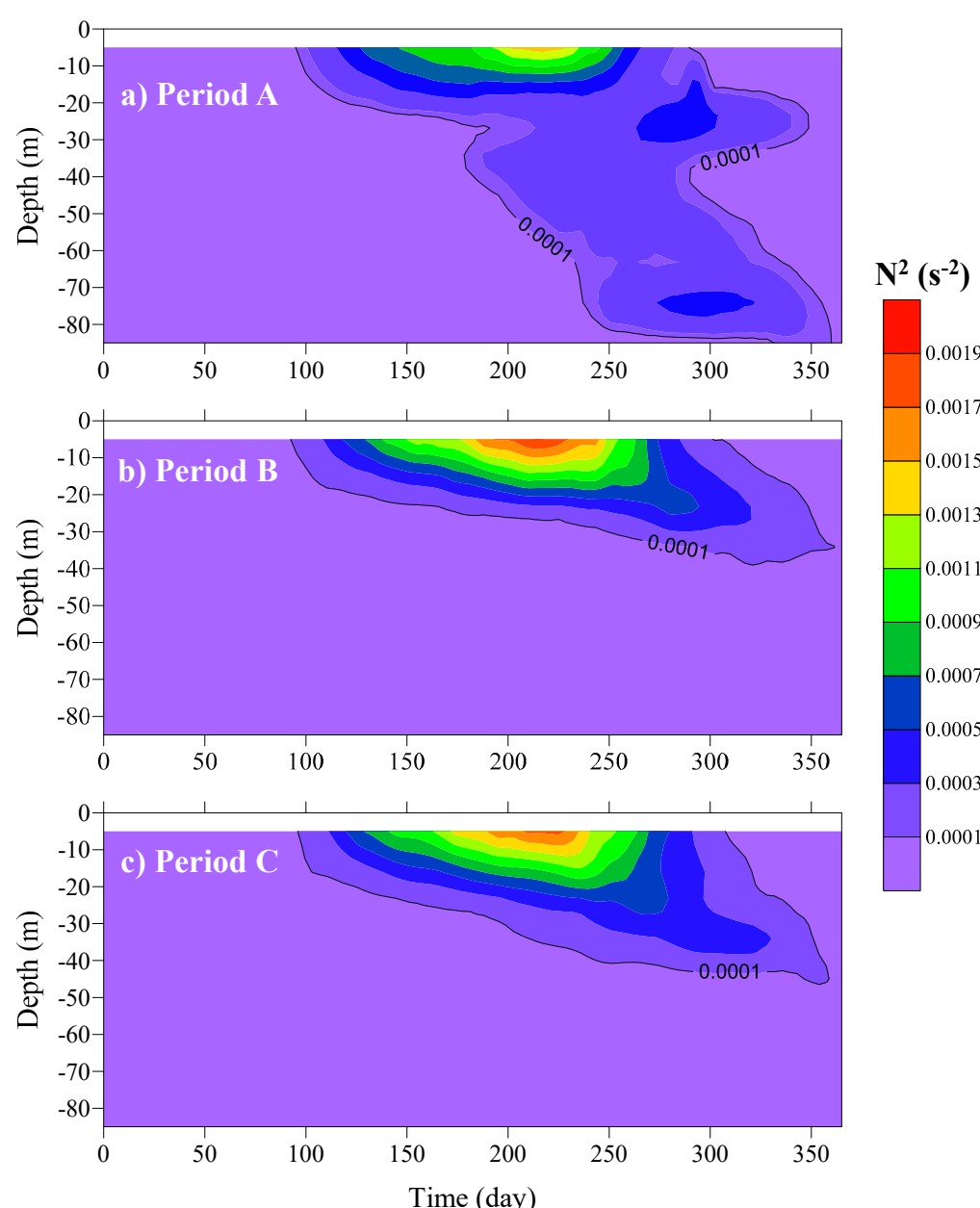

**Figure 8.** Contour plots of average $N^2$ for the three periods (**a**–**c**).

## 4. Discussion

### 4.1. Correlation Between Climate Forcing and Reservoir Temperatures

Pearson correlation coefficients ($r$) between the climate forcing and reservoir temperatures were obtained. Long-term correlation (1959–2016) was made only between climate forcing and SWT, excluding the 10- to 70-m temperatures. On the other hand, all water temperatures (0 to 70 m) were included in the short-term correlation for every period.

For the long term, SWT shows a positive correlation with air temperature for the summer half-year ($r$ = 0.73) and a negative one with wind speed ($r$ = −0.29) for the winter half-year. Meanwhile, other meteorological parameters acquire very weak correlations for both seasons. It has to be noted that strong correlation is not evidence of a mechanistic effect, i.e., correlation is not causation. The long-term correlation established specifically between air temperature and SWT should not be confused with the long-term trends generated

from the M–K test. While there is a strong positive correlation between the two parameters during the summer half-year, the M–K test showed no significant long-term trends in the same period. In the same way, while the correlation between the two parameters appeared to be weak during the winter half-year, the M-K test showed significant inverse trends.

For the short term, SWTs for Periods A and B show positive correlation with air temperature for April to September ($r_A$ = 0.76; $r_B$ = 0.86) and October to March ($r_A$ =0.64; $r_B$ = 0.73), with Period C exhibiting very weak correlation. The warming of the surface water occurs via downward longwave radiation, as associated with the increase in air temperature. SWT in Period B has a negative correlation with wind speed, specifically during the summer half-year ($r_B$ = −0.38). Furthermore, a weak correlation between SWT and basin-averaged rainfall is seen for Periods A and B for both seasons, but SWT is negatively correlated with basin-averaged rainfall during the winter half-year ($r_C$ = −0.58) for Period C. Considering the deeper layer of the reservoir, air temperature and wind speed have very low correlation with the temperatures at the 10- to 70-m depths. However, a positive correlation is detected between basin-averaged rainfall and the 70-m layer temperature, with *r* of 0.58 and 0.65 for Period A during the summer half-year and winter half-year, respectively.

To highlight the results for the short term, a negative correlation is found to exist between rainfall and deep-layer water temperature in Period A (summer half-year and winter half-year), between wind speed and SWT in Period B (summer half-year), and between rainfall and SWT in Period C (winter half-year). This correlation test results only suggest that within an individual short-term period, where a facility is operated, a certain climate forcing is associated with the temperature drop at a certain layer of the reservoir.

### 4.2. Effect of Facilities on Reservoir's Thermal Structure

The downstream outflow control by DPW and SW and the upstream inflow interception by VCs can explain the significant differences in water temperature distributions between the three periods for both the summer half-year and winter half-year, as shown in Figure 9.

In Period A, the river water largely disperses and remains for a longer period within the reservoir and replaces the colder water in the deep zone. This subsequent interaction of river water with the hypolimnetic water could explain the high correlation in Period A between bottom water temperatures and rainfall, wherein the latter serves as the primary source for river inflow. On the other hand, in Period B, the abstraction of water through SW leads the inflow to follow a narrow path along the upper layer and encourages most of the river water to be released directly out of the dam. The apparent thermocline formation during Period B not only limits the thermal advection between the upper and lower layers of the reservoir [23,29] but further shields the radiant heat transfer from the atmosphere to the hypolimnion [14]. This can further reaffirm why air temperature is highly correlated with SWT but not with the temperatures at the deeper layers. Furthermore, shallow withdrawals can diminish the internal heat in the reservoir over the summer and can offset the effect of further warming [52].

Meanwhile, during Period C, the curtains facilitated the plunging of the river water underneath it, limiting the flow to a layer way below the level of inflow and making a slightly wider epilimnion than in Period B. The two curtains acted as hydraulic and thermal barriers against the direct intrusion of the river water into the upper layer of the reservoir. River water has relatively larger velocity and different temperatures than the reservoir, and its manner of dispersion in the reservoir, as affected by the VC, essentially influences the thermal structure of the water body [23].

The average of PEA for each period strongly corresponds with the documented average trends of Schmidt's Stability Index (SSI) in the same reservoir [23]. Shallower withdrawals result in larger average values of PEA and SSI compared to deeper withdrawals. Likewise, Figure 8, which provides the reservoir's stability, as defined by $N^2$, shows that shallower withdrawals in Periods B and C produced stronger thermoclines, specifically

during the peak of heating in the year. The thermocline in Period A, on the other hand, appeared to be weaker and migrated deeper into the reservoir as the weather cooled down in autumn and winter. As mentioned earlier, surface releases strengthen the stratification while bottom releases induce the warming of the reservoir body, hence the lower thermal stability [27–29].

Looking at Figure 5, higher heat was stored in the reservoir during Period A compared to Periods B and C. In the case of the two latter periods, the heat exchange between the epilimnion and hypolimnion was strongly limited by the thermocline, hence the lower values of Q. In one study that carried out a heat budget analysis of the Sau Reservoir, it was found that hypolimnetic withdrawals increased the reservoir's annual Birgean heat budget (ABHB) while intermediate withdrawals produced stronger thermoclines and decreased both Q and ABHB [21]. Eventually, that study concluded that hydraulic management can partially counteract the effects of climate warming.

This present study strongly establishes that the different behavior of the thermal structure of the reservoir in the three periods is mainly caused by management and not climate warming and that some management strategies can be used to mitigate certain climate impacts. Nevertheless, this only generalizes the effects of surface releases with the use of the SW facility but has not yet explored the effect of withdrawals at different depths. The operation of the Ogouchi Reservoir can still be optimized in order to maintain desirable water quality not only in the main reservoir body but also with the released water downstream.

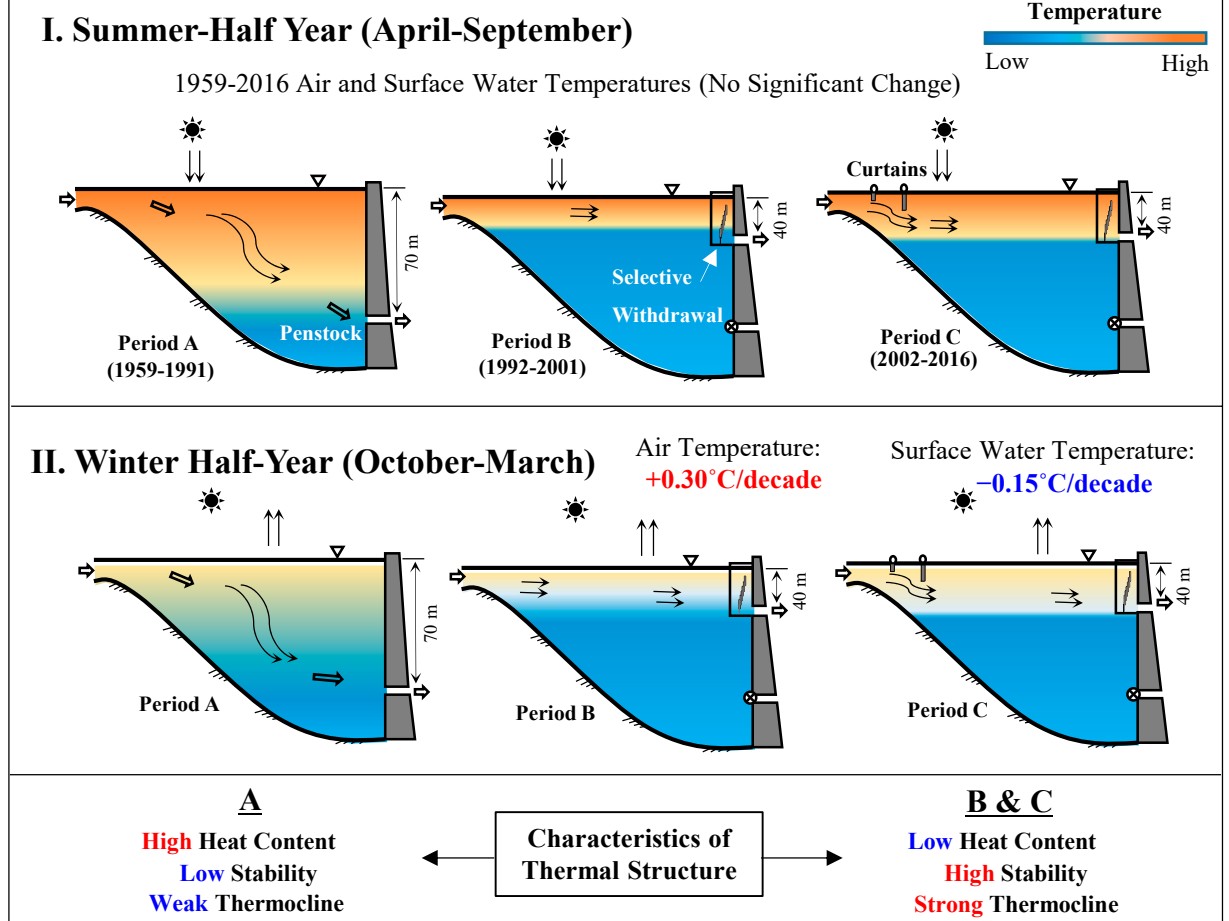

**Figure 9.** Effect of climate warming and facilities on the reservoir's thermal structure.

## 5. Conclusions

The effects of climate and varying facility operations on the temperature and thermal structure of a warm monomictic reservoir are evaluated in this study. Although air temperatures were rising, SWTs were found to be decreasing in the long term. Climate forcing affects the reservoir temperatures within the individual periods, but the varying reservoir operation has been identified to ultimately influence the differences in thermal responses among the periods. Kruskal-Wallis tests affirmed that the distributions of water temperatures were significantly different between the three periods, while rank-sum tests proved that Periods B and C were significantly different (colder) to Period A. The two latter periods exhibited lower heat content due to their shallower epilimnion but higher stability in terms of PEA due to stronger vertical stratification. The stratification, as defined by $N^2$, showed that Periods B and C developed stronger thermoclines than Period A. Flow interception by VCs upstream and outflow control by DPW and SW downstream play a large role in either inhibiting or enhancing the radiant heat transfer from the atmosphere to the reservoir and advection between epilimnion and hypolimnion with the presence of thermocline. This study reveals that the thermal condition of this reservoir is not significantly affected by the climate warming that was prominent during the colder seasons. Furthermore, reservoir operation bears a stronger influence on the temperature and thermal structure of the reservoir than climate change itself. The use of SW and VCs appears to be a promising key to mitigate the thermal impacts of climate warming. Future studies will include numerical simulations to determine the sensitivity of the reservoir's temperature and thermal structure with varying hydrometeorological parameters, assuming each facility is operated over the long term. Furthermore, the effects of management options on downstream temperatures can be studied through numerical simulation in the hope of recommending the most favorable operating measures for SW.

**Author Contributions:** Conceptualization, M.A.D. and K.Y.; Methodology, M.A.D.; Formal Analysis, M.A.D.; Writing—Original Draft, M.A.D.; Visualization, M.A.D.; Validation, T.S. and K.Y.; Writing—Review and Editing, T.S. and K.Y.; Supervision, T.S. and K.Y.; Resources, K.Y.; and Project Administration, K.Y. All authors have read and agreed to the published version of the manuscript.

**Funding:** This research received no external funding.

**Institutional Review Board Statement:** Not applicable.

**Informed Consent Statement:** Not applicable.

**Data Availability Statement:** This is a 3rd Party Data (Tokyo Metropolitan Government). Access to the data is highly restricted.

**Acknowledgments:** The authors are indebted to the Bureau of Waterworks of the Tokyo Metropolitan Government for the use of data. The valuable advice of Jonathan David Lasco is highly acknowledged for the statistical approaches in this study. The authors also thank Charles John Gunay for the arrangement of the meteorological data. The first author extends his sincere gratitude to the Tokyo Human Resources Fund for City Diplomacy for the grant for his Ph.D studies at the Tokyo Metropolitan University.

**Conflicts of Interest:** The authors declare no conflict of interest.

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
