# Peer review of "Mediating the Effects of Climate on the Temperature and Thermal Structure of a Monomictic Reservoir through Use of Hydraulic Facilities"

_water, doi:10.3390/w13081128_

Round 1

Reviewer 1 Report

Summary

This paper investigates how the thermal structure of a reservoir is influenced by selective withdrawal and by vertical curtains at the inlet. The investigated question is relevant in light of potential options for reservoir management to mitigate climate change impacts, and the available dataset is excellent for the purpose. However, I think the following points should be considered.

Major comments

1) The previously available literature is insufficiently considered. The authors state that literature on responses of reservoirs to climate warming are “very scarce”, but there are many other studies besides the few that are cited in the manuscript (Lewis et al, 2019, Helfer et al, 2012, Mi et al, 2019, Zouabi-Aloui, 2015), and which, except Lewis et al. (2019) are also not discussed in the introduction. There are also numerous studies on the effects of selective withdrawal on the thermal structure of reservoirs and on the downstream river temperature. Here is a list of a few relatively recent studies on the effects of either climate change or selective withdrawal on reservoir thermal structure, but many others can be found in the reference lists of those or elsewhere:

Azadi et al (2018): https://doi.org/10.1007/s11269-018-2109-z

Chang et al. (2015): https://doi.org/10.3390/w7041687

Zhang et al (2018): https://doi.org/10.1016/j.watres.2015.02.052

Kobler et al. (2018) https://doi.org/10.1007/s10584-018-2340-x and Kobler et al. (2019) https://doi.org/10.1002/hyp.13519

Feldbauer et al. (2020): https://doi.org/10.1186/s12302-020-00324-7

Rheinheimer et al. (2015), https://doi.org/10.1061/(ASCE)WR.1943-5452.0000447

Weber et al (2017): https://doi.org/10.1016/j.jenvman.2017.03.020

I would not expect a comprehensive review of all studies on these topics but the major previous findings and the novelty of the present study compared to previous investigations should be clearly presented in the introduction, and the new findings should be related to those from previous studies in the discussion.

2) I think the analysis of the data and the following discussion could gain a lot from presenting the averages and ranges of the seasonal dynamics of some properties (especially temperatures at different depths and the total heat content of the reservoir) and comparing them for the three different management periods. Since weekly measurements are available for the entire period, the dataset would be perfect to do that, by calculating averages and ranges for each week of the year and plotting them against the day of the year. Maybe the first year of each period should be skipped for such an analysis as the data seems to indicate some transitional effects in Figure 6.

3) Conversely, I think the long-term trends of different properties should be given less weight in the paper. Maybe most of the trend analysis could even be moved to a supplement. This is not the relevant part of the study, and it dilutes the main messages. As the authors state themselves, the durations of periods B and C are too short for calculating robust trends, furthermore, these might be further influenced by the transitional effects mentioned above. In my opinion, the long-term trends should be used only for two purposes in this paper: (i) to show that the different behaviour of the reservoir in the three periods is mainly caused by the management and not by climate warming, and (ii) to support the notion that the management strategies could be used to mitigate some climate impacts.

4) The effects of the management options on downstream temperatures are only mentioned in the introduction. However, it would also be important to discuss these effects further. The seasonal course of the temperature of the withdrawn water could easily be estimated from the available information for the three different periods. Ideally, if any data on this is available, it could then be related to the inflow temperature regime, which in the absence of upstream reservoirs, can be considered to be close to natural. Furthermore, the introduction only mentions that deep withdrawal could lead to too cold temperatures downstream, but also the opposite is possible in case of epilimnetic withdrawal. Olden and Naiman (2010) actually discuss both cases.

Minor comments

Line 38: it seems odd to group evaporation and bacterial activity together.

Line 49: The causal link (“Because...) between the withdrawal depths and the residence times is not correct in my opinion, the residence time as a first approximation only depends on the reservoir/lake volume and the total inflows.

Line 54: I don’t think that reservoirs are necessarily less sensitive to climate change than lakes, and I also don’t think that Adrian at al. (2009) claim this.

Line 68: Another important reason for selective withdrawal that could be mentioned here is to avoid low oxygen concentrations downstream.

Figure 1: As they are located within the selected rectangle, it would be useful to show the river gauging stations and the inflowing streams also on the reservoir map inset. The title “Legend:” could be removed.

Line 133: incomplete sentence

Table 1: please add elevation information for the meteorological stations.

Figure 2: Does this figure show the data from the dam station? Or is this average data for all stations listed in Table 1? If the latter, I think it would probably be better to show data only from one station rather than data from different stations for different parameters. But in any case, this should be specified in the caption.

Lines 152 ff: is there any reason to use different methods for calculating long-term (Sen’s slope) and short-term (linear regression) trends?

Line 163: “warming and cooling stage” are rather unusual terms, and don’t make too much sense, because the weather is already significantly cooling during what is still called the warming stage and vice versa. Why not simply use “summer half year” and “winter half year” or just specify the ranges (Oct-Mar, Apr-Sep)?

Equation (1): In this equation, according to rLakeAnalyzer, there should be T, not delta_T.

Equation (3): I don’t think an average stability N2 over 30 m is really meaningful. The main purpose of using N2 is to detect the strength of stratification at a certain depth, not over a large depth range. Averaging it over 30 m is in fact equal to just taking the density difference between 0 and 30 m depth. If that is what you would like to do, then preferentially use the so called “relative thermal resistance to mixing” which is a measure for the density difference between the epilimnion and the hypolimnion. Or just use that density difference directly.

Equation (4): W is the Wedderburn number, and the friction velocity is usually calculated from the average wind speed. What was the rationale for using the maximum windspeed instead, which maximum was used, and what was the equation to calculate u_star from windspeed?

Climate Trends: It seems pretty surprising to me that for entire Japan the winter air temperature trends are almost equal to those of the entire year, whereas for this specific catchment, the winter trends are much higher than the annual mean trends. Is there any explanation for this, or confirmation from other regional studies?

Figure 3: I don’t think calculating climate trends for short periods of 10 years is meaningful, see major comment (2) above.

Line 254: Preferably don’t use percentage changes for temperature. Temperature is not an absolute scale, the percentage change would be different if you use a different unit (Kelvin or Fahrenheit).

Figure 6: There must be something wrong here either in the numbers or in the unit. Is this the thermal energy per m2? The total energy content of the reservoir should be several orders of magnitude higher.

Line 390: For the same reason as in the comment to line 254, don’t use ratios of temperature.

Line 396: This is not generally true, surface temperature of many lakes exceeds mean air temperature also in summer.

Reviewer 2 Report

I think this is an excellent study.  There seems to be a little inconsistency with the quality of the writing, but they are minor (listed below).  The only drawback is that there is no consideration of seasonal climate forcings (eg. El Nino etc), but this may not be important in the location of the reservoir under study.  

Abstract

The length of operation for each period and average air temperature increase should be reported in the abstract.

I suggest you change “Reservoir operations were progressively changing into three distinct 15 periods namely” to “Reservoir operations were progressively changed into three distinct periods:”

This sentence:

“ Climate forcings affect the reservoir temperatures within individual periods but the varying operation is identified to influence ultimately the differences in thermal responses among periods.” is very ambiguous, after several readings of it, it became apparent that the following sentence appears to be saying the same thing, but in a much clearer way.  I suggest you remove the first sentence.

Line numbers

39: “The rise in SWT could result to:” should read “The rise in SWT could result in:”

Remove the “in”

41: ice covers should be ice cover

64: Literatures should be literature

66: on cold.. should be of cold..

79: the example of the VC on the river mouth seems to be conflated with its use in a reservoir situation.  I suggest this sentence needs to be rewritten to clarify.

130-131:   This sentence seems incomplete.  Additionally the type of analysis should be stated.

135: what site is being referred to?

137: replace ‘while the’ with ‘and’.  Remove “With this,” at the start of the next sentence

145: higher winds not larger winds.

156-157: this sentence belongs with the previous paragraph and some explanation of the reason for the different analysis approach between long-term and short-term trends.

184: reaches 30m depth.

200: the word ‘justified’ does not appear to be correct in this context.  Perhaps evidenced?

208: instead of the term ‘normal range of fluctuations’, I suggest you use a ‘stationary rainfall climate’. 

212: I’m not sure what a time category is – perhaps the authors mean different time periods?  Climate drivers (such as El Nino) of these periods need to be tested or at least considered in the discussion.

311: does the author mean a strong correlation?

339: there appears to be an extra space between ‘disperses’ and ‘and’

376 – 378: This is a definitive statement that requires some explanation.

408: on should be over

432: CRediT!

Reviewer 3 Report

The paper presents an analysis of water temperature data measured between 1958 and  2016 at the Ogouchi reservoir, aimed at disentangling the role of different management strategies on the stratification and vertical distribution of the water temperature. Although quite similar to previous works by the same Authors on the same Ogouchi reservoir, the work presents some new analyses and is quite interesting.

It should be noted that the study is based on a single measuring station for the water temperature (measures at different depths), which limits the knowledge of the spatial (i.e., mainly horizontal) distribution of the water temperature. The availability of remote sensed temperature data for the free surface (e.g., Pivato et al., 2020) could be useful shed light on the temperature spatial distribution of the upper layer and on the role played by three tributaries. At least, this issue should be discussed in the text.

The English needs to be improved. Some sentences are quite odd (e.g., lines 55-57 at page 2, but there are many other similar occurrences). For example, “forcing” is rarely used in plural form by native speakers.

Pivato, M., Carniello, L., Viero, D. P., Soranzo, C., Defina, A., & Silvestri, S. (2020). Remote sensing for optimal estimation of water temperature dynamics in shallow tidal environments. Remote Sensing, 12(1), 51. https://doi.org/10.3390/rs12010051

Round 2

Reviewer 1 Report

The authors did a very good job revising the paper and responded appropriately to most of the comments. Specifically, I think the new Figure 5 and potentially Figure 8 (but see below) significantly facilitate understanding what is going on in the reservoir. Nevertheless, I think the following points should be considered before publication.

Line 42: delete “and as such, most climate change-related studies have been established for this water body as affirmed by the plentiful papers mentioned above.”

Climate trends (sections 2.2 and 3.1): It has now been clarified that the data presented in Figure 2 are from the dam station. However, it is still unclear for most of the results presented whether they are results from individual stations or averages from multiple stations. If all the presented results are from the dam station, why are the other stations introduced at all? And if not, which are resulting from other stations?

Terms “heating stage” and “cooling stage”: I would like to insist that the terms “heating stage” and “cooling stage” are not appropriate. The terms imply that the periods are defined by the time when the reservoir is taking up heat or releasing heat, but that is not the case. They are defined by air temperature being above or below the annual average. I can understand the arguments of the authors against my previous suggestions (although April to September is the normal definition of summer half year), but then other terms should be found. Based on the argumentation of the authors, I would propose “warm season” and “cool season”, which does not have the issue of being different between hemispheres.

Equation (4): please change formatting, as it is formatted now it looks like E is a variable. Preferably use the same formatting for exponents as in Table 3.

Wedderburn number: I still have some issues with how the Wedderburn number is used here. First of all, I think the statement “The use of maximum wind speed was necessary to determine the possible maximum upwelling in the reservoir. Using the average wind speed for this purpose will not produce any meaningful result” is not ideal. Preferably the fact that the Wedderburn number should be used as a measure for the maximum upwelling should be introduced before describing how it is calculated. But if that is the aim, I think the Wedderburn number should then first be calculated for each individual observation (including also the variation of density difference with time) and then the lowest Wedderburn number should be used rather than the Wedderburn number for the day with the highest wind speed. Finally, maximum upwelling does not really make sense for the cold season, as this includes a period of homothermal conditions where delta rho and thus the Wedderburn number are both close to zero per definition.

Figure 8: I like the seasonal contour plots you added in general, but the data presented here is wrong. N2 is highest in the thermocline and low in the epilimnion and the hypolimnion.

Section 4.1: I am fine with the authors’ decision to keep the discussion on the climate trends, although I would have preferred to shorten or even remove it. However, if it is retained, then some points should be amended:

  • Lines 396 to 404: It seems obvious to me that the negative long-term trend is a result of the lower cool season temperatures (Table 3) due to the changed management procedures. The paragraph here rather confuses the reader in this respect. If it should be checked whether climate variables could have played an additional role, this should be done by comparing the mean values of the climate variables between the periods A, B and C, rather than the trends of these variables.
  • Line 407: The main effect of air temperature on surface water temperature is not via convection but via downward longwave radiation.

Finally, as pointed out by the other two reviewers for the original version, the English language could still be improved. In particular, there are some issues with the usage of plurals and singulars, as well as with using articles. In the following a few suggestions for sentences where I think the language quality distorts the meaning:

  • Line 44: change to “A number of recent literatures investigated the responses of reservoirs to climate change.”
  • Line 64: change to “The two types of inland water bodies therefore generally have considerably different temperature dynamics and respond differently to climate.”
  • Line 294: change to “Although the SW facility was in operation in both these two latter periods, the effect of VC may have played a role in the higher heat content of C.”
